# MSWI Bottom Ash Application to Resist Sulfate Attack on Concrete

**Yongzhen Cheng \*, Yun Dong, Jiakang Diao, Guoying Zhang, Chao Chen and Danxi Wu**

Faculty of Architecture and Civil Engineering, Huaiyin Institute of Technology, Huai'an 223001, China;
dyunhyit@hyit.edu.cn (Y.D.); 201861228007@njtech.edu.cn (J.D.); 18351888629@163.com (G.Z.);
201861228005@njtech.edu.cn (C.C.); 201861228010@njtech.edu.cn (D.W.)

\* Correspondence: 230139226@seu.edu.cn

**Abstract:** This research provides a strategy for partially replacing cement with municipal solid waste incineration (MSWI) bottom ash (BA) to improve the performance of concrete against sulphate attack. Mortar strength tests were performed firstly to evaluate the hydration activity of the ground BA. Concrete specimens were cured in standard conditions and immersed in a solution that contained 10% sodium sulfate. Then, the compressive strength of these specimens was measured to investigate the mechanical properties and durability of the concrete. Next, the capillary porosity of the concrete was determined from the volume fractions of water lost in specimens. Finally, the transport of the sulphate solution in concrete was analyzed using capillary rise, crystallization rate, and solution absorption tests. The results indicated that BA had a certain hydration activity. The equivalent replacement of cement by BA decreased the compressive strength of the specimens but increased the durability of the concrete. There was an excellent correlation between capillary rise height, sulfate solution absorption amount, crystallization rate, and coarse capillary porosity. The addition of BA can decrease the coarse capillary porosity and further slow the capillary transport and crystallization of sulfate solution in concrete. Overall, the replacement of cement with BA can improve the durability of concrete and actualize the utilization of MSWI residues as a resource.

**Keywords:** MSWI bottom ash; concrete; sulfate attack; capillary transport; crystallization

## 1. Introduction

The quantity of municipal solid waste continues to increase; it exceeded 2.1 billion tons as of 2017 in China, but it is still growing at a rate of 5% to 8% a year [1]. At present, incineration is the most effective way to realize the reduction and reutilization of municipal solid waste and render it harmless [2–4]. Incineration decreases waste quality by 70% and volume by up to 90% [5]. Thus, the purpose of waste reduction has been initially achieved. In addition, it generates energy from thermal combustion [6]. Waste incineration produces a large amount of bottom ash and fly ash. Bottom ash (BA) refers to the residue discharged from the end of the hearth, which is the main component of ash residue and is close to 80% of the total weight of bottom ash and fly ash. Therefore, further harmless treatment of municipal solid waste incineration (MSWI) bottom ash and its utilization as a resource are still urgent problems to be solved.

Cement concrete is the best-known type of construction material. Due to its good construction performance, economy, and durability, cement concrete has widely been used in structures such as buildings, bridges, and tunnels [7]. However, it has been found that concrete does not last as long as expected. Many destructive factors, such as sulphate attack, chloride ion penetration, carbonation, and more, can decrease the durability of concrete [8–10]. The reasons for damage to the durability of concrete due to sulfate attack are divided into two major categories, namely, physical attack and

chemical corrosion [11]. Chemical corrosion is mainly caused by chemical reaction between the salt solution and the hydration products of the cement [12]. Physical attack refers to the destruction of concrete by the crystallization of the salt solution [13]. This destruction comes from volume expansion after the salt crystallization. The resistance of concrete to sulfate attack is mainly affected by the content of tricalcium aluminate, the amount of cement, and its compactness. The addition of industrial mineral admixtures can effectively improve the performance of concrete against sulphate attack [14–16]. After cement is partly replaced by coal ash, silica fume, and slag, and the amount of cement in the concrete is reduced accordingly, the content of calcium aluminate is relatively reduced. Furthermore, compared with cement, such industrial mineral admixtures have lower fineness and particle size; therefore, concrete admixed with these admixtures has greater compactness. Theoretically, concrete mixed with a mineral admixture has better resistance to sulfate attack.

In this study, a strategy was proposed to improve the sulfate resistance of concrete by mixing it with MSWI bottom ash. This proposal was also based on the chemical constitution of MSWI bottom ash, which is typically rich in silica and calcium oxide with minimum amounts of heavy metals, as it is classified as nonhazardous waste by the China Hazardous Waste List. Consequently, it is available to be reused as a secondary building material [17–19]. Many studies have strongly stated that MSWI bottom ash has some hydration activity and can be used to manufacture mortar [20,21]. However, the use of MSWI bottom ash to resist sulfate attack on concrete has been rarely reported, and the transportation and crystallization processes of salt solution in concrete remain unknown. Herein, detailed research was conducted by means of strength analysis, porosity measurements, and capillary rise and crystallization tests. Various water/cement ratios (W/C) and feedings (cement/BA) were found to have great impacts on the mechanical properties and salt solution transportation.

## 2. Materials and Methods

### 2.1. Materials

#### 2.1.1. MSWI Bottom Ash

The MSWI bottom ash used in this research was sampled from a waste incineration power plant in Huai'an, China. The impurities and heavy metal ions in the samples were removed by washing treatment and magnetic separation. Ball-milling was conducted on an XQM-8 variable-frequency planetary ball mill to reduce the particle size to a maximum of 180 μm. Figure 1 shows the original and ground MSWI bottom ash. The particle size distribution of the BA was determined using a Bettersize 2000 laser particle analyzer, and the result is presented in Figure 2. Meanwhile a larger specific surface area of 6631 cm$^2$/g was obtained.

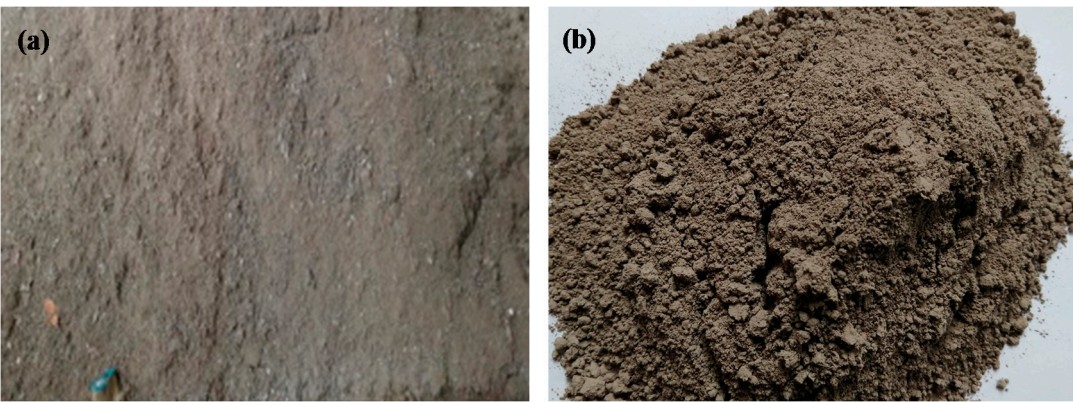

**Figure 1.** Municipal solid waste incineration (MSWI) bottom ash before (**a**) and after (**b**) ball milling.

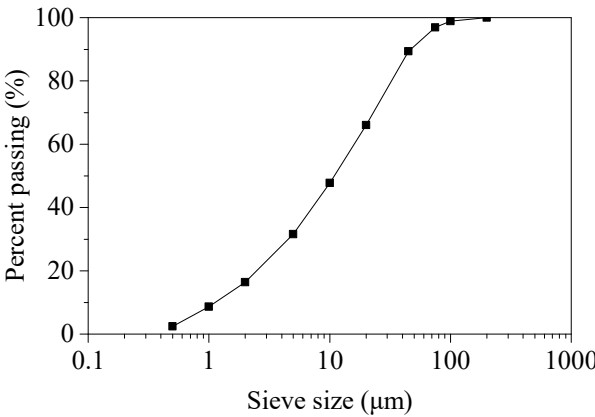

**Figure 2.** Particle size distribution for bottom ash (BA).

X-ray fluorescence spectroscopy (XRF) was performed using a Philips PW2400 instrument to determine the chemical composition of the MSWI bottom ash. The major, minor, and trace elements in the BA are presented in Table 1. Si, Al, Fe, and Ca together accounted for 68.59% of the total elements. Thus, the BA had a similar chemical composition to Portland cement. It belongs to a typical chemical system of $CaO$–$SiO_2$–$Al_2O_3$–$Fe_2O_3$ and should have hydration activity.

**Table 1.** The chemical (oxide) composition of BA.

| Oxides | wt % |
|--------|------|
| $SiO_2$ | 48.41 |
| CaO | 14.78 |
| $Al_2O_3$ | 11.99 |
| $Na_2O$ | 3.25 |
| $Fe_2O_3$ | 5.40 |
| $SO_3$ | 1.86 |
| $K_2O$ | 1.42 |
| MgO | 1.78 |
| $TiO_2$ | 0.76 |

### 2.1.2. Cement

Ordinary Portland cement with a grade of 42.5 was used in this study. The physical and mechanical properties of the cement measured in accordance with the Chinese standards are presented in Table 2.

**Table 2.** Material properties of cement.

| Property | Requirements | | Test Result | Test Method |
|----------|--------------|---|-------------|-------------|
| Soundness | Qualified | GB 175-2007 [22] | Qualified | T0505-2005 (JTG E30-2005) [23] |
| Initial setting time (min) | $\not< 45$ | | 97 | T0505-2005 (JTG E30-2005) [23] |
| Final setting time (min) | $\not> 390$ | | 180 | |
| Compressive strength (MPa) | 3d $\not< 17$ | | 28.6 | T0506-2005 (JTG E30-2005) [23] |
| | 28d $\not< 42.5$ | | 57.6 | |
| Flexural strength (MPa) | 3d $\not< 3.5$ | | 4.5 | |
| | 28d $\not< 6.5$ | | 9.0 | |
| Specific area ($cm^2$/g) | 3000 | | 3550 | T 050-2005 (JTG E30-2005) [23] |

### 2.1.3. Natural Sand and Crushed Stone

The natural sand and crushed stone used in this research were sampled from a construction site in Huai'an, China. Sieve testing was performed in accordance with JCJ 52-2006 [24]. The grain size distribution curves of the natural sand and crushed stone are presented in Figures 3 and 4,

respectively. The flat–elongated particle content, crushing value, clay content, and other parameters were determined in accordance with Chinese standards, and the test results are presented in Table 3.

The water used for manufacturing the mortar and concrete in this research was local tap water.

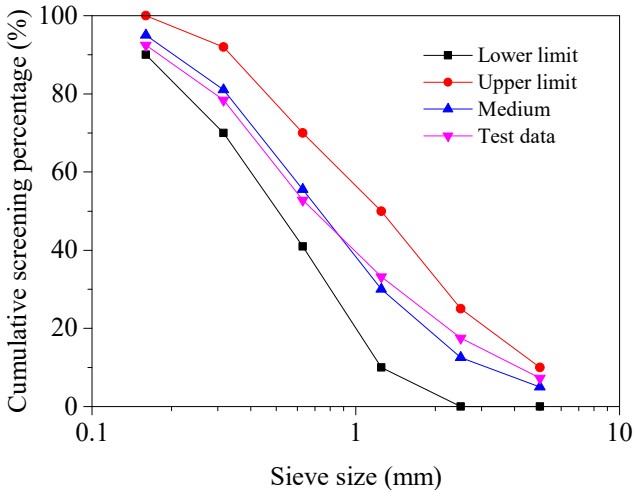

**Figure 3.** Grain size distribution curve of natural sand.

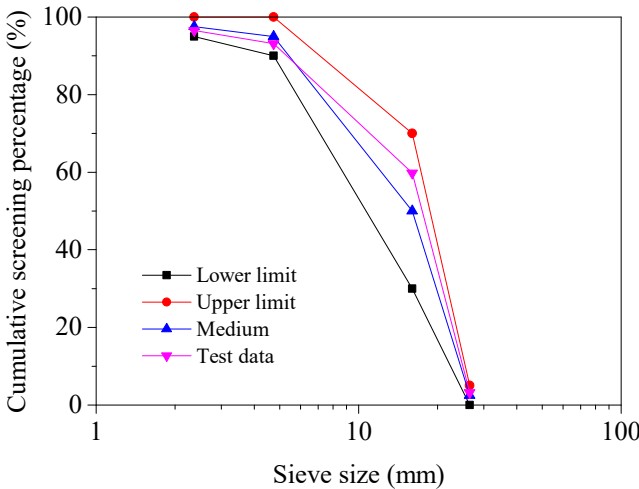

**Figure 4.** Grain size distribution curve of crushed stone.

**Table 3.** Material properties of natural sand and crushed stone.

| Property | Natural Sand | | Crushed Stone | | Standard |
|---|---|---|---|---|---|
| | Requirements | Test Result | Requirements | Test Result | |
| Fineness modulus | NA | 2.57 | — | — | JCJ 52-2006 [24] |
| Flat–elongated particles (wt %) | — | — | $\ngtr15$ | 13.5 | |
| Clay content (wt %) | $\ngtr3.0$ | 2.5 | $\ngtr1.0$ | 0.8 | |
| Clay lump content (wt %) | $\ngtr1.0$ | 0.8 | $\ngtr0.5$ | 0.3 | |
| Crushing value (wt %) | — | — | $\ngtr16$ | 12.8 | |
| Ruggedness (wt %) | $\ngtr8$ | 6 | $\ngtr8$ | 5 | |
| Apparent density (g/cm$^3$) | NA | 2.586 | NA | 2.714 | |

### 2.1.4. Mix Proportion Design

The mix proportion of mortar, determined according to GB/T 17671-199 [25], is shown in Table 4. The mix ratio of materials in mortar should be one cement, three standard sand, and one-half water by weight. Here, standard sand refers to natural quartz sea sand with $SiO_2$ content greater than 96%. After washing and sieving, it was processed into standard sand that meets the ISO requirements.

Concrete with a strength grade of C30 was prepared in the laboratory. The mixture ratio design of the concrete was carried out in accordance with JGJ 55-2000 [26]. Firstly, the mixed strength of the concrete was determined according to the strength grade of the designed concrete. Then, the water–cement ratio (W/C) was calculated based on the mixed strength of the concrete and strength value of the cement. Next, according to the slump and particle size of the stone, the water consumption was determined, and the amount of cementing material was calculated. Finally, the sand ratio was selected, and the amount of sand and crushed stone was calculated according to the maximum particle size of the stone and fineness modulus of the sand.

The polycarboxylic water-reducing agent mixed into the concrete accounted for 0.8% of the cementing material by volume. The water reducing rate was up to 29%. The final concrete mix proportions are shown in Table 4.

**Table 4.** Mix proportions of materials in the mortar and concrete.

| Material | Cement Mortar | Cement Concrete |
|---|---|---|
| Cement (kg/m$^3$) | 450 | 330 |
| Water (kg/m$^3$) | 225 | 130 |
| Natural sand (kg/m$^3$) | 1350 | 614 |
| Crushed stone (kg/m$^3$) | — | 1248 |
| Water-cement ratio (W/C) | 0.5 | 0.4 |

### 2.2. Combination Scheme

Laboratory tests were performed to evaluate the effect of BA content on concrete under sulphate attack. In addition, strength tests were conducted on the mortar specimens to evaluate the hydration activity of BA. The combination scheme of cementitious materials for mortar and concrete is presented in Table 5. In this study, five groups of mortar and concrete samples with BA content from 10% to 30% were prepared. In addition, a group of samples without BA was prepared. Each group consisted of three specimens.

**Table 5.** Cementitious material combination scheme for mortar and concrete.

| Material | Cement (kg/m$^3$) | BA (kg/m$^3$) | Material | Cement (kg/m$^3$) | BA (kg/m$^3$) | Cement/BA | Designation |
|---|---|---|---|---|---|---|---|
| Cement mortar | 405 | 45 | Cement concrete | 297 | 33 | 90:10 | C90BA10 |
| | 382.5 | 67.5 | | 280.5 | 49.5 | 85:15 | C85BA15 |
| | 360 | 90 | | 264 | 66 | 80:20 | C80BA20 |
| | 337.5 | 112.5 | | 247.5 | 82.5 | 75:25 | C75BA25 |
| | 315 | 135 | | 231 | 99 | 70:30 | C70BA30 |

### 2.3. Experimental Methods

#### 2.3.1. Mechanical Property Measurements

The specimen preparation and strength tests of the mortar were conducted in accordance with GB/T 17671-199 [25]. The well-stirred mixtures of cement, BA, sand, and water were put into a mold (40 × 40 × 160 mm) fixed on a vibrating table; after vibrating 120 times, the mixtures together with the mold were stored in a curing room (maintained at 20 ± 2 °C and no less than 95% RH). Then, form stripping was carried out. After 28 days of curing in standard conditions, the flexural and compressive strengths of the mortar were determined using a bending and compression tester.

The activity index of BA was determined with reference to GB T1596-2017 [27]. According to this standard, the compressive strengths of mortar with 30% and without fly ash were tested under a water-to-binder ratio of 0.5. Here, the activity index is defined as the compressive strength of the

mortar with BA divided by the compressive strength of the mortar without BA, and it can be calculated as follows:

$$H = \frac{R}{R_0} \times 100 \tag{1}$$

where $H$ is the activity index (%); $R$ is the compressive strength of mortar with BA at 28 days of curing (MPa); and $R_0$ is the compressive strength of mortar without BA at 28 days of curing (MPa).

The concrete specimens were prepared and cured according to GB/T 50081-2002 [28]. All the materials were accurately weighed and put into the mixing pan. After stirring, specimens $100 \times 100 \times 100$ mm in size were manufactured by the vibration molding method. Next, all specimens were left to stand for 24 h at a temperature of $20 \pm 5$ °C. In order to evaluate the strength property of the concrete, the specimens were cured in a curing room (maintained at $20 \pm 2$ °C and no less than 95% RH). After 28 days of curing, the compressive strength of the concrete was determined using a WAW-B Electro-hydraulic universal testing machine.

For durability tests, specimens $100 \times 100 \times 100$ mm in size were divided into two series. The first one was cured under standard curing conditions. The second one was firstly cured at T = $20 \pm 2$ °C and RH $\geq$ 95% for 28 days and then totally immersed in a solution containing 10% sodium sulfate for 60 days; the solutions were renewed monthly. The sulfate damage was evaluated mechanically by determining the compressive strength loss of the specimens using the following equation:

$$Strength\ loss\ (\%) = \frac{R_1 - R_2}{R_1} \times 100 \tag{2}$$

where $R_1$ is the compressive strength of concrete specimens under standard curing conditions (MPa) and $R_2$ is the compressive strength of concrete specimens in sulfate solutions (MPa).

### 2.3.2. Concrete Porosity Measurements

The concrete porosity was determined in accordance with the standard test method [29]. The porosity of concrete can be obtained indirectly from the water loss rate of saturated concrete specimens under certain conditions. The concrete specimens were prepared and cured for 28 days in standard conditions. After vacuum saturation, the water on the specimen surface was wiped off using a dry cloth. The mass of the specimen, measured using an electronic balance, was noted down. Then the specimen was cured at RH = 90% for 30 days. When water diffusion in the concrete reached equilibrium states, the mass of the same specimen was measured again. Next, the specimen was oven-dried to a constant weight at T = 105 °C. The specimen was finally weighed after cooling. The concrete porosity was calculated using the following equations:

$$P_{fine} = \frac{(M_0 - M_1) \times \rho_C}{M_0 \times \rho_w} \times 100\% \tag{3}$$

$$P_{total} = \frac{(M_0 - M_2) \times \rho_C}{M_0 \times \rho_w} \times 100\% \tag{4}$$

$$P_{coarse} = P_{total} - P_{fine} \tag{5}$$

where $P_{total}$ is the total porosity of the concrete specimen (%); $P_{fine}$ is the fine capillary porosity of the concrete specimen (%); $P_{coarse}$ is the coarse capillary porosity of the concrete specimen (%); $M_0$ is the mass of the saturated concrete specimen (kg); $M_1$ is the mass of the concrete specimen cured at RH = 90% for 30 days (kg); $M_2$ is the mass of the dried concrete specimen; $\rho_c$ is the density of the concrete (kg/m$^3$); and $\rho_w$ is the density of the water (kg/m$^3$).

### 2.3.3. Capillary Rise and Crystallization Tests

A concrete specimen $150 \times 150 \times 150$ mm in size was prepared and cured by following the standard method. After 28 days of curing, a cylinder (100 mm in diameter and 150 mm in length)

was drilled from the specimen (Figure 5a). The cylinder part was oven-dried to a constant weight at T = 60 °C. Then, this part was partially immersed in water to measure the rising height of capillary water in the concrete; the test method is shown in Figure 5c. The hollow part was used to measure the capillary crystallization rate of sodium sulfate solution in the concrete. A schematic diagram of the capillary crystallization tests is shown in Figure 5b. The bottom surface was firstly sealed. Then, a solution containing 5% sodium sulfate was poured into the cavity. The crystallization of sodium sulfate solution from the side wall and the thickness of concrete wall were observed and recorded every half hour. The above tests were performed at T = 20 ± 2 °C and RH = 60% ± 5%. The capillary crystallization rate was calculated using the following equation:

$$V_s = L/T \tag{6}$$

where $V_s$ is the capillary crystallization rate (cm/s); $L$ is the thickness of the sidewall (cm); and $T$ is the time of original crystallization on the concrete surface (s).

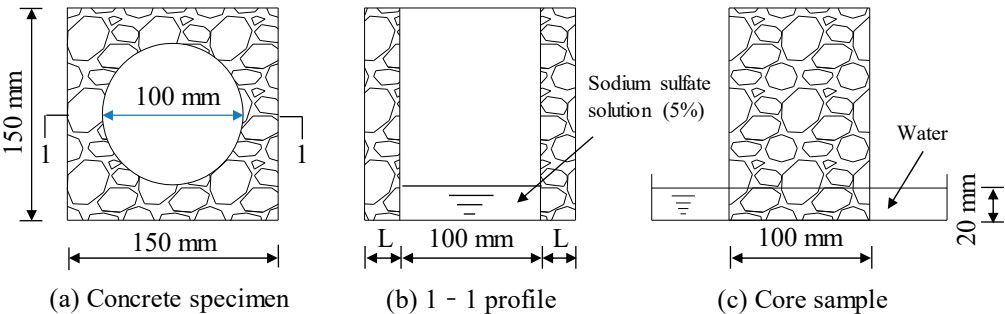

(a) Concrete specimen　　　(b) 1 - 1 profile　　　(c) Core sample

**Figure 5.** Schematic diagram of the capillary rise and crystallization tests (**a**) Concrete specimen without core sample; (**b**) 1-1 Profile of Figure 5a; (**c**) Core sample.

### 2.3.4. Solution Absorption Measurements

A specimen (100 mm in diameter and 100 mm in length) was prepared and cured for 28 days. Next, the solution absorption was measured in the laboratory; a schematic diagram is shown in Figure 6. All the side faces of the concrete specimen were sealed using epoxy resin. The specimen was oven-dried to a constant weight at T = 60 °C for no less than 12 h and was weighed after cooling. Then, the specimen was placed in a solution containing 5% sodium sulfate, keeping the water surface 5 mm above the bottom surface. The solution temperature was kept constant at 20 °C. The water on the specimen surface was wiped off using a dry cloth, and the weight of the specimen was measured at regular intervals.

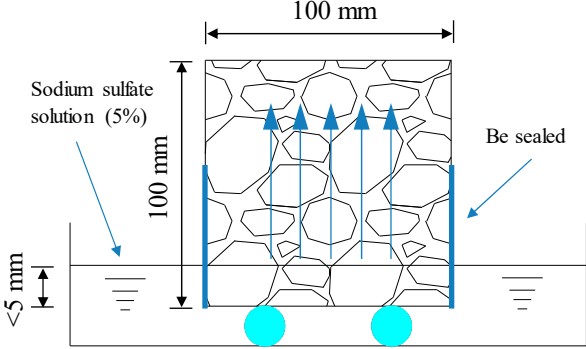

**Figure 6.** Schematic diagram of solution absorption measurements.

## 3. Results and Discussion

### 3.1. Strength Properties of Mortar and Concrete

The flexural and compressive strengths of the mortar samples with various contents of BA at 28 days of curing are shown in Table 6. Both the compressive and flexural strengths of the mortar decreased with increasing BA content, which is similar to the results of previous research [30–32]. The compressive strength of mortar showed a large decrease at the addition of BA from 10% to 20%. There was not a significant decrease in compressive strength at the addition of BA from 20% to 25%. Beyond 25%, the compressive strength decreased significantly again. Similar to the compressive strength, with increasing BA, the flexural strength also showed a rapid reduction first and then a gentle reduction, followed by another significant reduction.

Low hydration activity of BA was obtained, and the activity index was only 43% with the addition of 30% BA. The hydration activity of BA was apparently smaller in comparison with the records in the literature [33]. In the literature, a low W/C value of 0.38 was used. However, a similar industrial mineral admixture, electric arc furnace dust, has high hydration activity with W/C values from 0.35 to 0.7 [34]. Hence, the W/C of 0.5 used in this study is not the main cause of the low hydration activity. The samples in previous research were prepared by melting the MSWI fly ash at a high temperature and then water-quenching [33]. In this study, the BA was prepared by artificially removing the impurities. In addition, the particle size of the BA sample was controlled under 180 μm, making it hard to densify the microscopic structure of the mortar. In addition, our BA sample had a higher content of $SiO_2$ and a lower amount of CaO in comparison with the finer samples [35], and CaO may participate in the cement hydration process.

**Table 6.** Flexural strength and compressive strength of mortar samples with various contents of BA.

| Samples | Flexural Strength (MPa) | | Compressive Strength (MPa) | | Activity Index (%) |
|---|---|---|---|---|---|
| | Average Value | Standard Deviation | Average Value | Standard Deviation | |
| Without BA | 9.0 | 0.8 | 57.1 | 2.9 | — |
| C90BA10 | 5.2 | 0.3 | 34.6 | 2.6 | 61 |
| C85BA15 | 5.0 | 0.2 | 32.0 | 2.7 | 56 |
| C80BA20 | 5.0 | 0.5 | 27.6 | 2.2 | 48 |
| C75BA25 | 4.8 | 0.4 | 26.8 | 2.5 | 47 |
| C70BA30 | 4.5 | 0.3 | 24.4 | 2.1 | 43 |

Figure 7 shows the compressive strength values of the concrete samples with different contents of BA at 28 days of curing. The compressive strength of the concrete decreased gradually with increasing addition of BA. In addition, the water/cement ratio (W/C) had a significant influence on the compressive strength: the larger the W/C, the smaller the compressive strength. The concrete samples with the addition of 10%, 15%, and 20% BA at a W/C of 0.35 met the strength requirement of C30, at 37 MPa, 34 MPa, and 32 MPa, respectively. When the W/C was 0.40, only the concrete samples with the addition of 10% and 15% BA met the strength requirement, at 35 MPa and 31 MPa, respectively. Unfortunately, the compressive strengths of all concrete specimens at a W/C of 0.35 were less than 30 MPa. This shows that the addition of BA could not improve the strength performance of the concrete due to its relatively low hydration activity.

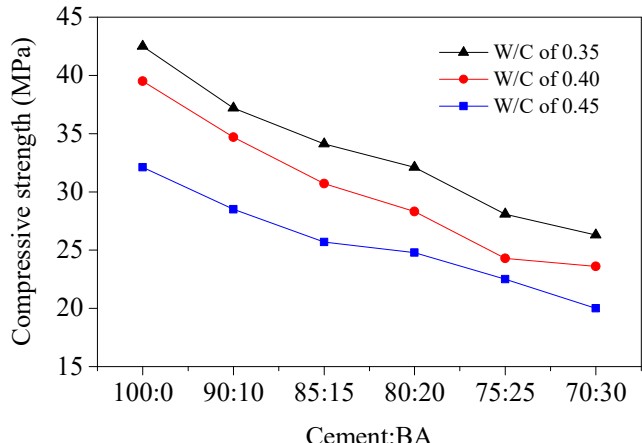

**Figure 7.** Influence of BA content on the compressive strength of concrete.

### 3.2. Concrete Durability

Figure 8 shows the compressive strength values of concrete specimens cured in standard conditions and immersed in sulfate solutions, also giving the strength loss. All the concrete specimens were manufactured with a W/C of 0.4. The compressive strengths of the concrete samples with the addition of 10%, 20%, and 30% BA decreased from 30.5 MPa, 27.7 MPa, and 26.0 MPa to 25.7 MPa, 25.6 MPa, and 24.9 MPa, respectively. Meanwhile the strength loss decreased from 15.6% to 7.5%, then to 4.2%. Thus, the more BA is added to the concrete, the smaller the strength loss will be. The BA can thus improve the performance of concrete against sulphate attack. To some extent, this is due to the very high surface area of the analyzed BA, which can fill in the pores of the concrete and prevent the sulphate solution from seeping into concrete. A similar phenomenon was found in cement concrete mixed with fly ash and other recycled micro powders [36–38].

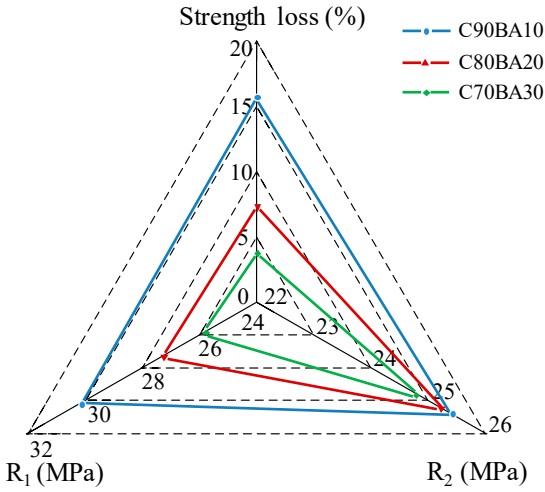

**Figure 8.** Strength loss in $Na_2SO_4$ solution.

The porosity values of cement concrete samples at different W/C and addition levels of BA are shown in Figure 9. The total porosity, coarse capillary porosity, and fine capillary porosity all increased with increasing W/C. The addition of BA was able to efficiently decrease the porosity of the concrete, especially the coarse capillary porosity. The pore structure and porosity are the key factors that affect the strength of cement-based materials [39]. They are also the decisive factors in the resistance of cement-based materials to invasive media [40]. The attack resistance of cement-based materials is poor when the porosity is large and the pores are interconnected. On the contrary, better

performance of concrete against attack can be obtained. This explains why the concrete with BA had better sulfate resistance.

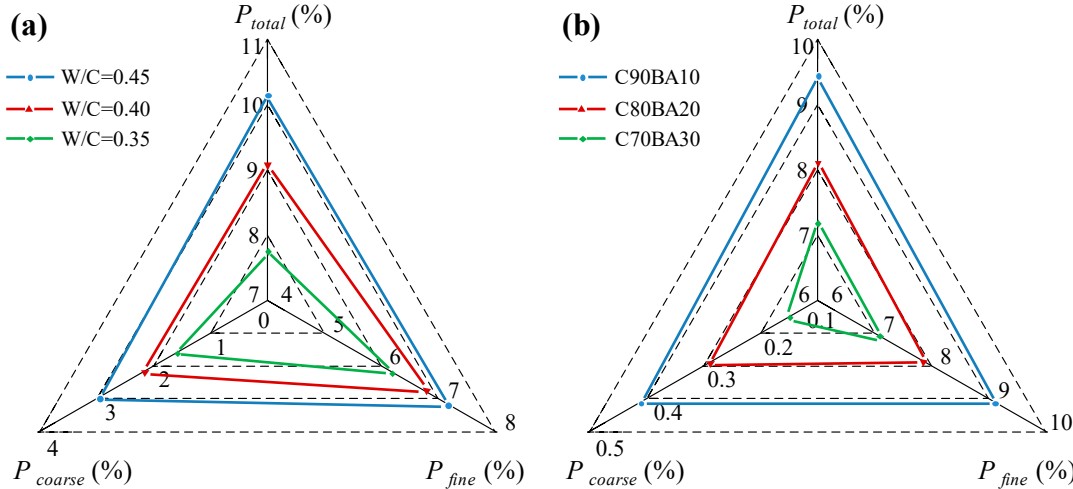

**Figure 9.** Influence of (**a**) W/C and (**b**) BA content on porosity of cement concrete.

### 3.3. Intrusion of Concrete by Sodium Sulfate Solution

Figure 10 shows the heights of capillary rise with time at different W/C values and BA contents. The height of capillary rise increased with increasing W/C. The capillary rise height of the concrete with BA was obviously lower than that of the concrete without BA.

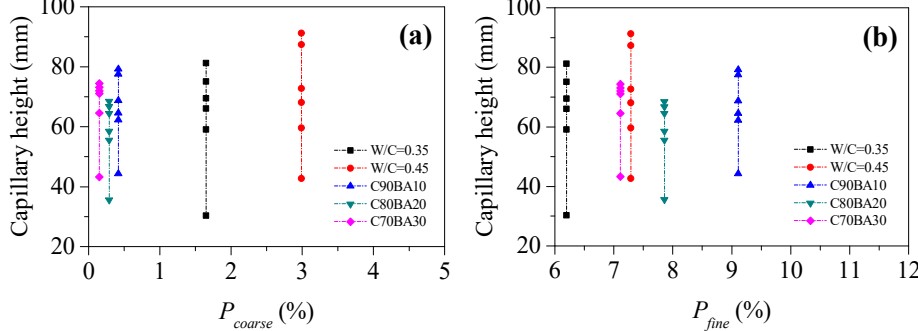

**Figure 10.** Height of capillary rise at (**a**) coarse and (**b**) fine capillary porosity.

The correlations of the capillary rise height versus the coarse and fine capillary porosity are shown in Figure 11. There was a good correlation between the height of capillary rise and the coarse capillary porosity (pore size of ≥30 nm) of the concrete, with a correlation coefficient of 0.819. However, the height of the capillary rise had little correlation with the fine capillary porosity (pore size of <30 nm), with a correlation coefficient of only 0.0669. Thus, the capillary porosity types with various pore sizes have different effects on capillary rise in concrete. The coarse capillary porosity plays a key role in the capillary transport of a solution in concrete [41].

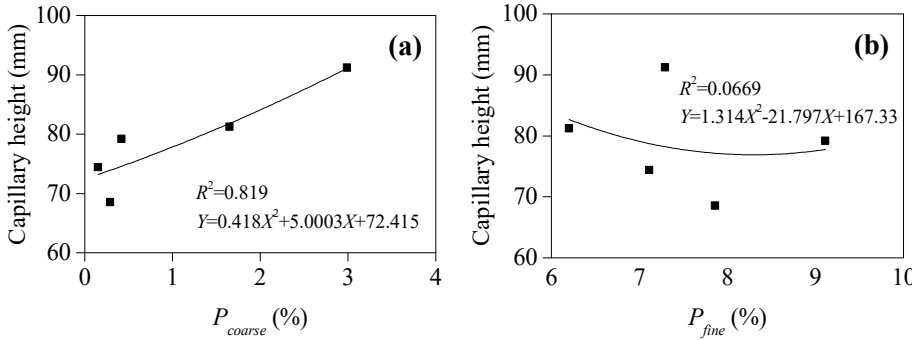

**Figure 11.** Correlations of capillary rise height vs. (**a**) coarse and (**b**) fine porosity.

Figure 12 shows the correlations of the crystallization rate of capillary transmission versus the coarse and fine porosity. The crystallization rate of capillary transmission increased with increasing W/C. A shorter time for sodium sulfate solution to reach the surface of concrete results in a larger area of crystallization on the concrete surface. The crystallization rate of the concrete with BA decreased significantly. In addition, the crystallization rate decreased with increasing BA content. There was a good correlation between the crystallization rate of capillary transmission and the coarse capillary porosity, with a correlation coefficient of 0.959. However, the crystallization rate had little correlation with the fine capillary porosity, with a correlation coefficient of only 0.0973. That also fully explained the importance of the coarse capillary porosity in the capillary transport of a solution in concrete.

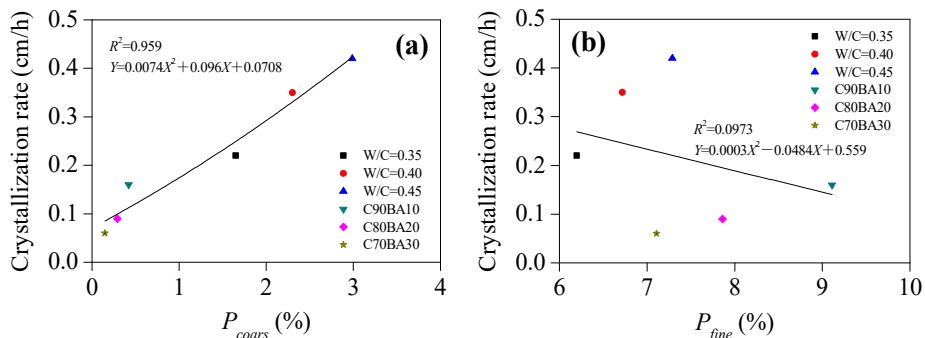

**Figure 12.** Correlations of crystallization rate vs. (**a**) coarse and (**b**) fine porosity.

The influence of the coarse and fine porosity on the mass of sodium sulfate solution absorption and their correlations are shown in Figure 13. W/C had a significant influence on the mass of sodium sulfate solution absorption, which was largest at a W/C of 0.45, followed by a W/C of 0.35. After the addition of BA, the absorption amount of sodium sulfate solution in the concrete decreased significantly, and it deceased with increasing BA content. There was a good correlation between the absorption amount of sodium sulfate solution and the coarse capillary porosity, with a correlation coefficient of 0.9793. However, a poor correlation between the absorption amount of sodium sulfate solution and the fine capillary porosity was found.

In conclusion, the fine capillary porosity has little influence on the height of capillary rise, sodium sulfate solution absorption, and crystallization rate of capillary transmission, while the coarse capillary porosity plays a key role in the intrusion of sodium sulfate solution into concrete. The concrete samples at various W/C values and BA contents had different porosity properties. Thus, W/C and BA content have an obvious influence on the transfer and crystallization of salt solution in concrete. The height of capillary rise, sodium sulfate solution absorption, and crystallization rate decreased with increasing BA content. All this proves that the addition of BA can improve the performance of concrete against sulphate attack.

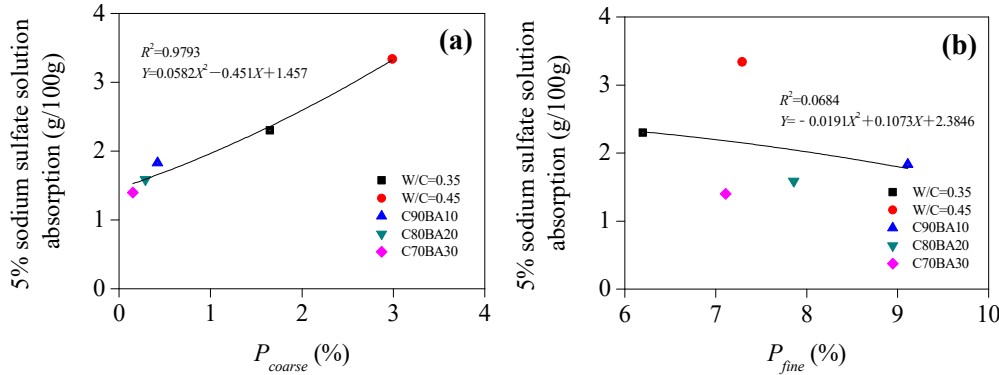

**Figure 13.** Correlations of mass of sodium sulfate solution absorption vs. (**a**) coarse and (**b**) fine porosity.

## 4. Conclusions

In this research, the durability of concrete with added BA was evaluated by immersing specimens in a solution containing 10% sodium sulfate. Then, capillary rise, crystallization rate, and solution absorption tests were performed to investigate the intrusion of concrete by sodium sulfate solution. Moreover, the crucial type of porosity was found with the help of porosity measurements. The following conclusions can be drawn:

(1) BA has some hydration activity due to its chemical constitution. Even if 30% of the cement is replaced with BA, the compressive and flexural strength values are still greater than 42% and 50%, respectively, of those of the mortar without BA.

(2) The compressive strength of the concrete decreased with increasing BA content; this is due to the weak pozzolanic reactivity of BA. However, BA can improve the performance of concrete against sulphate attack because BA has a very high surface area and fills in the pores of concrete.

(3) The coarse capillary porosity plays a key role in the capillary transport and crystallization of sulfate solution in concrete.

(4) The W/C and BA content have a certain impact on the porosity of concrete and further affect the capillary height, absorption amount, and crystallization rate of sulfate solution in concrete.

Furthermore, due to BA adjusting the strength and durability of the concrete, the optimum range of BA in concrete is from 10% to 15%. In this range, BA can not only improve the durability of concrete but also ensure the strength of concrete to meet design requirements.

The deficiency of this study is that no suitable treatment measures were found to enhance the hydration activity of BA. The objective of future research studies in this area is to improve the activity of BA by wet and high-temperature treatment. On this basis, the effect of BA on the strength and durability of concrete can be studied.

**Author Contributions:** Y.C. proposed the research method for analyzing the intrusion of sodium sulfate solution to reveal the durability of concrete. The experiments were performed and the manuscript was written by J.D., G.Z., C.C. and D.W. Y.D. proposed revision suggestions for this manuscript.

**Funding:** This research was funded by the Construction System Science and Technology Project in Jiangsu, China (Grant No. 2018ZD045), and the Training Program on Entrepreneurship and Innovation for University Students (Grant No. 2019JGYJ003 and 201911049023Y).

**Conflicts of Interest:** The authors declare that there are no conflicts of interest regarding the publication of this paper.

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
