# Peer review of "MSWI Bottom Ash Application to Resist Sulfate Attack on Concrete"

_applsci, doi:10.3390/app9235091_

Round 1
Reviewer 1 Report
The paper presents a clear and well explained experimental campaign aimed at studying both the activity of MSWI BA in mortars and concrete durability against sulfate attack. There are few modifications required:
> Lines 38-39: it is preferable to state that sulfate attack is one among the main causes of concrete deterioration, together with carbonation and chlorides-induced corrosion.
> Line 49-51: recall that increased fineness is present only in industrial mineral admixtures, instead when dealing with heterogeneous, recycled materials, this might not happen, reducing the activity of the material.
> Tables 2 and 3: check the symbols (might be a problem of visualization building the pdf file)
> Table 6: possibly add the standard deviation of the results
> Table 6: explaining why the activity index is so low, authors may would to consider that activity (as well as cementing coefficient or k-value for SCMs) are influenced by w/c ratio, and thus the choice of using a relatively high w/c compared for instance to that used in ref.32 might cause have values with less magnitude than those found in literature (see da Silva Magalhaes 2017).
> Figure 7: not clear how many concrete batches were tested, comparing the description provided in Section 2.2.
Author Response
Our responses:
Thanks for this reviewer’s efforts in reviewing our manuscript. The reviewer’s comments help us to improve the manuscript.
Lines 38-39: The cause of concrete deterioration is re-stated as follows: However, it has been found that concrete does not last as long as the expectation. Many destructive factors, such as sulphate attack, chloride ion penetration, carbonation and more, can decrease the durability of concrete. Line 49-51: The sentence is rewritten as follows: Furthermore, compared with cement, such industrial mineral admixtures have less fineness and particle size, therefore concrete admixed with admixture has greater compactness. Theoretically, concrete mixed with mineral admixture has better resistance to sulfate attack. The symbols in Tables 2 and 3 have been checked. Here the symbols “≯” and “≮” means “not more than” and “not less than”. We have added the standard deviation of the results in Table 6. The low activity index of BA has been explained as follows: The low hydration activity of BA is obtained, and the activity index is only 43% at the addition of 30% BA. The hydration activity of BA is apparently smaller in comparison with the record of literature [33]. In this literature, a low W/C of 0.38 was used. However, the similar industrial mineral admixtures, electric arc furnace dust has the high hydration activity with W/C from 0.35 to 0.7 [34]. Hence, the W/C of 0.5 used in this study is not the main cause of low hydration activity. The samples were prepared by melting the MSWI fly ash at a high temperature and then water-quenched in previous researches [33]. In this study, the BA were prepared by artificially removing the impurities. In addition, the particle size of BA sample is controlled under 180 μm, which is hard to make the microscopic structure of mortar denser. In addition, it has a higher content of SiO2 and a lower amount of CaO in comparison with the finer samples [35], in which CaO may participate in the cement hydration process. We have described the concrete batches in Section 2.2.
Sincerely,
Yongzhen Cheng
Nov. 18, 2019

Reviewer 2 Report
In my opinion this is a very interesting work. The authors study the possibility of applying Bottom ash of municipal solid waste incineration (BA-MSWI) for the manufacture of concrete and mortars, and studying the resistance to sulfate attack in concrete.
Therefore, two series of material, one of mortars where BA are used replacing between 90 and 70% of cement by these waste in both, mortar series and concrete series. The authors studied mechanical behaviour, porosity, capillarity, and in the end, resistance to sulfate attack tests were performed.
Dosages and mechanical strength results are adequate and consistent with previous explanations, the porosity is also consistent with these studies, correlations are established between the capillarity of the different components of the concrete and finally correlation between the sodium sulfate solutions absorbed by the concretes and the mortars, in relation to their components.
In my opinion, this document is appropriate, and I propose to improve the conclusions section. I consider that the authors have to include their opinions regarding the optimal substitutions that should be applied, and they should include a final conclusion after the specific ones, indicating the general benefit of this investigation and possible future researches.
Author Response
Our responses:
Thanks for this reviewer’s efforts in reviewing our manuscript. The reviewer’s comments help us to improve the manuscript.
We have added opinions regarding the optimal substitutions in the conclusions section and stated as follows: Furthermore, by adjusting the strength and durability of the concrete with BA, optimum range of BA in concrete is from 10% to 15%. In this range, BA can not only improve the durability of concrete, but also ensure the strength of concrete to meet the design requirements. We have added research prospects in the conclusions section and stated as follows: The deficiency of this study is that no suitable treatment measures have been found to enhance the hydration activity of BA. The objective of future research studies in this area is to improve the activity of BA by wet and high temperature treatment. On this basis, the effect of BA on the strength and durability of concrete is studied.
Sincerely,
Yongzhen Cheng
Nov. 18, 2019
